# A Bayesian approach to breaking things: efficiently predicting and repairing failure modes via sampling

**Charles Dawson**
Department of Aeronautics and Astronautics
MIT United States
cbd@mit.edu

**Chuchu Fan**
Department of Aeronautics and Astronautics
MIT United States
chuchu@mit.edu

**Abstract:** Before autonomous systems can be deployed in safety-critical applications, we must be able to understand and verify the safety of these systems. For cases where the risk or cost of real-world testing is prohibitive, we propose a simulation-based framework for a) predicting ways in which an autonomous system is likely to fail and b) automatically adjusting the system's design to preemptively mitigate those failures. We frame this problem through the lens of approximate Bayesian inference and use differentiable simulation for efficient failure case prediction and repair. We apply our approach on a range of robotics and control problems, including optimizing search patterns for robot swarms and reducing the severity of outages in power transmission networks. Compared to optimization-based falsification techniques, our method predicts a more diverse, representative set of failure modes, and we also find that our use of differentiable simulation yields solutions that have up to 10x lower cost and requires up to 2x fewer iterations to converge relative to gradient-free techniques. Accompanying code and video can be found at https://mit-realm.github.io/breaking-things/.

**Keywords:** Automatic design tools, root-cause failure analysis, optimization-as-inference

## 1  Introduction

From power grids to transportation and logistics systems, autonomous systems play a central, and often safety-critical, role in modern life. Even as these systems grow more complex and ubiquitous, we have already observed failures in autonomous systems like autonomous vehicles and power networks resulting in the loss of human life [1]. Given this context, it is important that we be able to verify the safety of autonomous systems *prior* to deployment; for instance, by understanding the different ways in which a system might fail and proposing repair strategies.

Human designers often use their knowledge of likely failure modes to guide the design process; indeed, systematically assessing the risks of different failures and developing repair strategies is an important part of the systems engineering process [2]. However, as autonomous systems grow more complex, it becomes increasingly difficult for human engineers to manually predict likely failures.

In this paper, we propose an automated framework for predicting, and then repairing, failure modes in complex autonomous systems. Our effort builds on a large body of work on testing and verification of autonomous systems, many of which focus on identifying failure modes or adversarial examples [3, 4, 5, 6, 7, 8], but we identify two major gaps in the state of the art. First, many existing methods [4, 5, 9, 7] use techniques like gradient descent to search *locally* for failure modes; however, in practice we are more interested in characterizing the distribution of potential failures, which requires a global perspective. Some methods exist that address this issue by taking a probabilistic approach to sample from an (unknown) distribution of failure modes [6, 10]. However, these methods suffer from a second major drawback: although they can help the designer predict a range of

7th Conference on Robot Learning (CoRL 2023), Atlanta, USA.

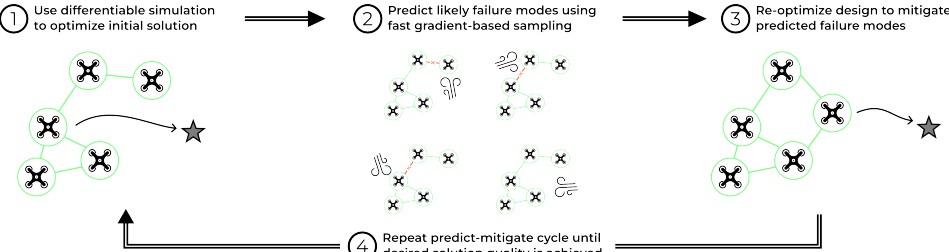

Figure 1: An overview of our method for predicting and repairing failure modes in autonomous systems, shown here handling connectivity failures in a drone swarm.

failure modes, they do not provide guidance on how those failure modes may be mitigated; they are also inefficient due to their use of gradient-free inference methods.

We address all of these drawbacks to develop a framework, shown in Fig. 1, for predicting and repairing failure modes in autonomous systems. Taking inspiration from inference-based methods [10, 6], we make three novel contributions:

1. We reframe the failure prediction problem as a probabilistic sampling problem, allowing us to avoid local minima and quickly find high likelihood, high severity failure modes.

2. We exploit the duality between failure prediction and repair to not only predict likely failure modes but also suggest low-cost repair strategies.

3. We employ automatic differentiation to take advantage of fast gradient-based sampling algorithms, substantially improving performance relative to the state of the art.

We demonstrate our approach on several large-scale robotics and control problems: swarm formation control with up to 10 agents, multi-robot search with up to 32 agents, an electric power transmission network with up to 57 nodes and 80 transmission lines. We compare our approach with baselines for both failure mode prediction and repair, showing that our framework outperforms the state-of-the-art and scales well beyond the capabilities of existing tools, converging to solutions that are up to 10x lower cost while requiring less than half as many iterations. We also demonstrate that the repair strategies developed using our approach can be deployed on hardware for the multi-robot search example, and a software implementation can be found at https://mit-realm.github.io/breaking-things/.

## 2   Related Work

**Model-based verification**   Early approaches to model-based verification and fault identification used symbolic logical models of the system under test to formally reason about failures using (computationally expensive) satisfiability (SAT) solvers or search [11, 12]. More recent approaches to model-based failure mode identification have used mathematical models of the system dynamics to frame the problem as a reachability [13] or optimal control [14] problem. The challenge in applying these methods is that it may be difficult or impossible to construct a symbolic model for the system under test. In this work, we seek to retain the interpretability of model-based techniques while eliminating the requirement for a fully symbolic model, using automatically differentiable computer programs instead. Such models are comparatively easy to construct [8] and can even include implicitly differentiable components such as the solutions to optimization problems [15].

**Adversarial testing**   Verification using adversarial optimization has been applied in both model-based [7, 5, 9] and model-free [3] contexts. Generally speaking, model-based adversarial techniques use gradient-based optimization to locally search for adversarial examples that cause a system failure, then use gradient-based optimization to locally repair those failures [7, 5]. The drawback of these methods is that they are inherently local and typically yield only a single adversarial coun-

terexample. Model-free approaches [3] can avoid the issue of local minima by using zero-order black-box optimization techniques but incur additional computational cost as a result. In contrast, we take a probabilistic approach where sample-efficient gradient-based sampling algorithms can be used to escape local minima and efficiently generate multiple potential failure cases [16].

**Inference** Ours is not the first work to take a probabilistic approach to failure mode prediction. O'Kelly *et al.* develop an end-to-end verification pipeline for autonomous vehicles based on adaptive importance sampling [10], and Zhou *et al.* develop a failure mode prediction system based on gradient-free Markov Chain Monte Carlo (MCMC) [6]. We take inspiration from these works and make two key improvements. First, these existing works focus exclusively on predicting likely failure modes — they do not include a method for mitigating these failure modes once discovered — while we combine failure mode prediction with repair by recognizing the duality between these problems. Second, we use differentiable simulation to replace inefficient zero-order MCMC algorithms with fast gradient-based samplers, resulting in a substantial performance improvement.

There is also a complimentary body of work on algorithms for rare-event simulation [17, 18] that provide extensions to MCMC-based sampling algorithms that perform well even when we seek to simulate extremely rare failure cases. Our framework is completely compatible with rare-event simulation strategies commonly used in Sequential Monte Carlo (SMC), and we view the incorporation of these methods into our framework as a promising direction for future work.

## 3 Assumptions and Problem Statement

At the heart of our approach is a simulation model of the system under test, parameterized by two distinct sets of parameters. The *design parameters* $x \in \mathcal{X} \subseteq \mathbb{R}^n$ are those parameters that the system designer may vary, while the *exogenous parameters* $y \in \mathcal{Y} \subseteq \mathbb{R}^m$ are those that may vary uncontrollably (due to environmental variation, adversarial disturbance, the actions of other agents, etc.). Together, $x$ and $y$ define the *behavior* of the system $\xi \in \Xi$ (e.g. a trace of all relevant states and observables) through the *simulator function*, denoted $\xi = S(x, y)$. In addition, we assume that a *cost function* $J(\xi)$ is known; i.e. $J$ reflects the property that the system designer wishes to verify. A summary of our notation is provided in Table 1 in the appendix; we will use "designs" and "failure modes" interchangeably with "design parameters" and "exogenous parameters", respectively.

**Assumption 1:** $S$ and $J$ are programs that we can automatically differentiate almost everywhere. This setting is more general than the case when an explicit mathematical model is known, but less general than a black-box setting (although many black-box systems in robotics can be automatically differentiated [19]). **Assumption 2:** $x$ and $y$ are continuous random variables with known, automatically differentiable, and potentially unnormalized prior probability densities $p_{x,0}(x)$ and $p_{y,0}(y)$. It may be counter-intuitive to model the design parameters as random variables, but this choice allows us to capture constraints on the design space by assigning low probability to infeasible designs. The prior distribution for $y$ can be either estimated from historical data or constructed to reflect constraints on the operational domain. We restrict our focus to the continuous-parameter case, but our approach can be extended to handle mixed discrete parameters using block-resampling [20].

In this context, *failure prediction* entails finding exogenous parameters $y^*$ that, for some given $x$, lead to a high cost. To ensure that predicted failures are plausible, we must also find values for $y^*$ with high prior likelihood. To achieve this balance, we define the metric of *risk-adjusted cost*

$$J_r(x, y) = J \circ S(x, y) + \log p_{y,0}(y) \tag{1}$$

where $\circ$ denotes function composition. Failure prediction is thus the problem of finding parameters $y^*$ that lead to a high risk-adjusted cost; moreover, since it is likely that $J_r$ will have multiple local minima with respect to $y$ (i.e. multiple likely failure modes), we wish to sample a set $\left\{y_1^*, \ldots, y_{n_y}^*\right\}$ of such failures. To generate this set, we replace deterministic optimization $y^* = \arg\min_y J_r(x, y)$ with sampling from the unnormalized *pseudo-posterior* (in the sense defined in [21]).

$$y^* \sim p(y^*|x) \propto p_{y,0}(y^*)e^{J \circ S(x,y^*)} \tag{2}$$

Likewise, the *failure repair* problem seeks to find design parameters $x^*$ that both have high prior likelihood (thus respecting the designer's prior beliefs about the design space) and result in a low cost across a range of anticipated failure modes; i.e. sampling from the unnormalized pseudo-posterior

$$x^* \sim p(x^*|y_1^*, \ldots, y_{n_y}^*) \propto p_{x,0}(x^*)e^{-\sum_i J \circ S(x^*, y_i^*)/n_y} \tag{3}$$

# 4 Approach: Adversarial Inference

The primary challenge in sampling from these failure and repair distributions is that they will naturally shift as the design changes. Once the design has been updated to account for the current set of predicted failures, those failures will likely be out of date. To address this issue, we define a novel adversarial sampling method to alternate between sampling improved failure modes $\{y_1^*, \ldots, y_n^*\}$ and then sampling improved design parameters $x^*$ to repair those failure modes, thus improving the robustness of the design while maintaining an up-to-date set of failure modes.

Our algorithm (detailed in Algorithm 1) proceeds in the style of a sequential Monte Carlo algorithm [18]. We begin by initializing $n_y$ potential failure modes and $n_x$ candidate designs sampled from their respective prior distributions. In each of $K$ rounds, we first sample $n_x$ new candidate designs from distribution (3) to repair the current set of predicted failure modes. We then select the design that performs best against all currently-predicted failures and sample $n_y$ new sets of exogenous parameters (each representing a potential failure mode) from distribution (2). To sample from distributions (2) and (3), we use $n_x$ and $n_y$ parallel executions of a Markov chain Monte Carlo (MCMC) sampler. In order to handle potential multimodality in the design and failure space, we include optional tempering to interpolate between the prior and target distributions [18].

Our proposed adversarial inference algorithm can accept any MCMC sampling algorithm as a subroutine, either gradient-free or gradient-based. In our experiments, we compare the results of using both gradient-free (random-walk Metropolis-Hastings, or RMH) and gradient-based (Metropolis-adjusted Langevin algorithm, or MALA [22]); both of these are included in the appendix. Empirically, gradient-based samplers typically converge faster, particularly on high-dimensional problems, but in cases where a differentiable simulator is not available, a gradient-free sampler will suffice. We use MCMC for the sampling subroutine in all of our experiments, but our framework is also compatible with other approximate inference methods (e.g. variational inference).

---

**Algorithm 1:** Failure prediction and repair using gradient-based sampling

**Input:** Population sizes $n_x$, $n_y$; rounds $K$; substeps $M$; stepsize $\tau$; tempering $\lambda_1, \ldots, \lambda_K$.

**Output:** Robust design $x^*$ and a set of failures $\left\{y_1^*, \ldots, y_{n_y}^*\right\}$ with high risk-adjusted cost.

1   Initialize candidate designs $[x]_0 = \{x_1, \ldots, x_{n_x}\}_0$ sampled from $p_{x,0}(x)$

2   Initialize candidate failures $[y]_0 = \{y_1, \ldots, y_{n_y}\}_0$ sampled from $p_{y,0}(y)$

3   **for** $i = 1, \ldots, K$ **do**

4      $p_{x,i}(x) := p_{x,0}(x)e^{-\lambda_k/n_y \sum_{y \in [y]_{i-1}} J \circ S(x,y)}$

5      $[x]_i \leftarrow \text{Sample}([x]_{i-1}, M, \tau, p_{x,i})$    ▷ Update candidate designs using predicted failures

6      $p_{y,i}(y) := p_{y,0}(y)e^{\lambda_k \min_{x \in [x]_{i-1}} J \circ S(x,y)}$    ▷ Update failure predictions for new best design

7      $[y]_i \leftarrow \text{Sample}([y]_{i-1}, K, \tau, p_{y,i})$

8   **return** $[y]_K$, $x^* = \arg\max_{x \in [x]_N} p_{x,i}(x)$        ▷ Choose best design

---

On a theoretical level, any MCMC sampler will be sound so long as the resulting Markov chain is ergodic and satisfies detailed balance [23]. Unfortunately, there can be a large gap between asymptotic theoretical guarantees and practical performance. First, if the target distribution is multimodal and the modes are well-separated, then MCMC algorithms may be slow to move between modes, yielding a biased sampling distribution. To mitigate this effect, we include a tempering schedule $0 \leq \lambda_1 \leq \ldots \leq \lambda_K \leq 1$ to interpolate between the prior and target distributions and run multiple MCMC instances in parallel from different initial conditions. Empirically, we find that tempering is not always needed, but we include it for completeness.

The second potential practical challenge arises from the continuity and differentiability (or lack thereof) of the simulator and cost function $J \circ S$. Although gradient-based MCMC samplers like MALA remain sound so long as the target distribution is continuously differentiable almost everywhere (i.e. discontinuous or non-differentiable on a set of measure zero), in practice performance may suffer when the target distribution has large discontinuities. Because of these issues, we design our method to be compatible with either gradient-based or gradient-free sampling algorithms, and we compare the results of using both methods in Section 6.

The final practical consideration is that although the stochasticity in our sampling-based approach can help us explore the design and failure spaces, we incur a degree of sub-optimality as a result. When using gradient-based sampling, we have the option to reduce this sub-optimality by "quenching" the solution: switching to simple gradient descent (e.g. using MALA for the first 90 rounds and then gradient descent on the last 10 rounds). In practice, we find that quenching can noticeably improve the final cost without compromising the diversity of predicted failure modes.

## 5    Theoretical Analysis

Our prediction-and-repair framework can work with both gradient-free or gradient-based sampling subroutines, but it is important to note that gradients, when available, often accelerate convergence. To support this observation, we provide non-asymptotic convergence guarantees for the gradient-based version of our algorithm, drawing on recent results in Ma et al. [16].

To make these guarantees, first assume that $J$ is $L$-Lipschitz smooth. Second, assume that the log prior distributions $\log p_{y,0}$ and $\log p_{x,0}$ are $m$-strongly convex outside a ball of finite radius $R$. The first assumption is hard to verify in general and does not hold in certain domains (e.g. rigid contact), but it is true for most of our experiments in Section 6. The second is easy to verify for common priors (e.g. Gaussian and smoothed uniform). Let $d = \max(\dim x, \dim y)$ be the dimension of the search space and $\epsilon \in (0, 1)$ be a convergence tolerance in total variation (TV) distance.

**Theorem 5.1.** *Consider Algorithm 1 with the stated assumptions on smoothness and log-concavity. If $m > L$ and $\tau = \widetilde{\mathcal{O}}\left((d \ln L/(m - L) + \ln 1/\epsilon)^{-1/2} d^{-1/2}\right)$, then sampling each round with TV error $\leq \epsilon$ requires at most $M \leq \widetilde{\mathcal{O}}\left(d^2 \ln \frac{1}{\epsilon}\right)$ steps.*

Since convergence time for each round of prediction and mitigation scales only polynomially with the dimension of the search space, our method is able to find more accurate failure predictions (and thus better design updates) than gradient-free methods with the same sample budget.

**Proof sketch**    Ma et al. [16] show that gradient-based MCMC enjoys fast convergence on non-convex likelihoods so long as the target likelihood is strongly log-concave in its tails (i.e. outside of a bounded region). It would be unrealistic to assume that the cost $J(x, y)$ is convex, but we can instead rely on the strong log-concavity of the prior to dominate sufficiently in the tails and regularize the cost landscape. A formal proof is included in the appendix.

## 6    Experimental Results

There are two questions that we must answer in this section: first, does reframing this problem from optimization to inference lead to better solutions (i.e. lower cost designs and predicted failures that accurately cover the range of possible failures)? Second, does gradient-based MCMC with differentiable-simulation offer any benefits over gradient-free MCMC when used in our approach?

We benchmark our algorithm on a range of robotics and industrial control problems. We compare against previously-published adversarial optimization methods [7, 5] and compare the results of using gradient-based and gradient-free MCMC subroutines in our approach. We then provide a demonstration using our method to solve a multi-robot planning problem in hardware. The code used for our experiments can be found at https://mit-realm.github.io/breaking-things/.

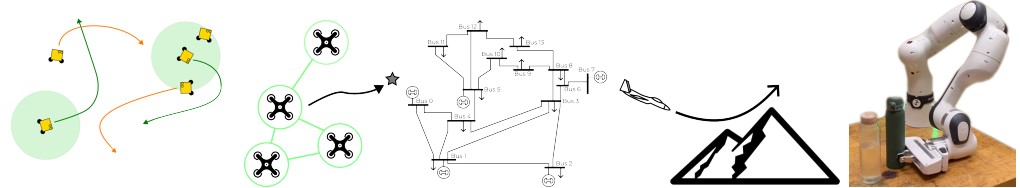

Figure 2: Environments used in our experiments. (Left to right) Multi-agent search-evasion, formation control, power dispatch, aircraft ground collision avoidance, and manipulation by pushing.

**Baselines** We compare with the following baselines. **DR**: solving the design optimization problem with domain randomization $\min_x \mathcal{E}_y[J_r(x, y)]$. **GD**: solving the adversarial optimization problem $\min_x \max_y J_r(x, y)$ by alternating between optimizing a population of $n_x$ designs and $n_y$ failure modes using local gradient descent, as in [7, 5, 24]. We also include two versions of our method, using both gradient-free (**RMH**) and gradient-based (**MALA**) MCMC subroutines. All methods are given the same information about the value and gradient (when needed) of the cost and prior likelihoods. The gradient-free version of our approach implements quenching for the last few rounds.

**Environments** We use three environments for our simulation studies, which are shown in Fig. 2 and described more fully in the appendix. **Multi-robot search:** a set of seeker robots must cover a search region to detect a set of hiders. $x$ and $y$ define trajectories for the seekers and hiders, respectively; failure occurs if any of the hiders escape detection. This environment has small (6 seeker vs. 10 hider, $\dim x = 60$, $\dim y = 100$) and large (12 seeker vs. 20 hider, $\dim x = 120$, $\dim y = 200$) versions. **Formation control:** a swarm of drones fly to a goal while maintaining full connectivity with a limited communication radius. $x$ defines trajectories for each robot in the swarm, while $y$ parameterizes an uncertain wind velocity field. Failure occurs when the second eigenvalue of the graph Laplacian is close to zero. This environment has small (5 agent, $\dim x = 30$, $\dim y = 1280$) and large (10 agent, $\dim x = 100$, $\dim y = 1280$) versions. **Power grid dispatch:** electric generators must be scheduled to ensure that the network satisfies voltage and maximum power constraints in the event of transmission line outages. $x$ specifies generator setpoints and $y$ specifies line admittances; failures occur when any of the voltage or power constraints are violated. This environment has small (14-bus, $\dim x = 32$, $\dim y = 20$) and large (57-bus, $\dim x = 98$, $\dim y = 80$) versions. **F16 GCAS:** a ground collision avoidance system (GCAS) must be designed to prevent a jet aircraft, modeled with aerodynamic effects and engine dynamics, from crashing into the ground. $x$ defines a control policy neural network ($\dim x \approx 1.8 \times 10^3$) and $y$ defines the initial conditions ($\dim y = 5$). **Pushing:** a robot manipulator must push an object out of the way to reach another object. Failure occurs if the object is knocked over while pushing. $x$ defines a planning policy network ($\dim x \approx 1.2 \times 10^3$) and $y$ defines the unknown inertial and frictional properties of the object being pushed, as well as measurement noises ($\dim y = 7$). We implement our method and all baselines in Python using JAX. All methods were run with the same population sizes and total sample budget, using hyperparameters given in the appendix.

**Solution quality** For each environment, we first solve for an optimized design and a set of predicted failure modes using each method. We then compare the performance of the optimal design on the predicted failure modes with the performance observed on a large test set of $10^5$ randomly sampled exogenous parameters. The results of this experiment are shown in Fig. 3.

We find that both DR and GD often fail to predict failure modes that accurately cover the tail of worst-case behaviors: in the formation and power grid examples, both DR and GD falsely indicate that all predicted failures have been successfully repaired, despite a long tail of possible failures in both cases. In the search example, adversarial GD is able to predict a set of useful failure modes, but DR fails to do so. Only our method (with both gradient-free and gradient-based MCMC) accurately predicts the worst-case performance of the optimized design.

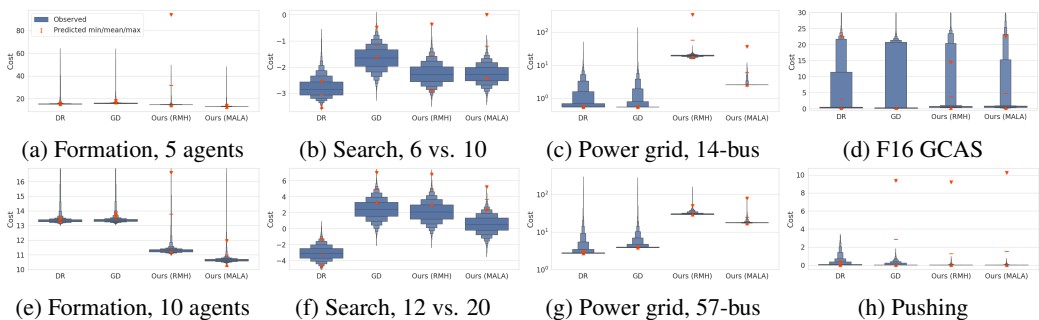

Figure 3: A comparison of the cost of the optimal design on the predicted failure modes (red) and $10^5$ randomly sampled test cases (blue).

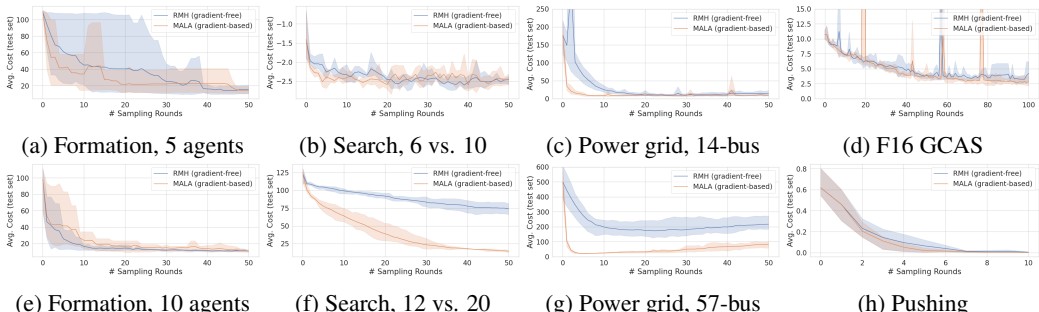

Figure 4: Convergence rates of gradient-based (orange) and gradient-free (blue) MCMC samplers when used as subroutines for Algorithm 1. Shaded areas show min/max range over 4 random seeds.

In addition to comparing the quality of the predicted failure modes, we can also compare the performance and robustness of the optimized design itself. On the search problem, our method finds designs with slightly improved performance relative to GD (but not relative to DR, since DR is not optimizing against a challenging set of predicted failure modes). On the formation problem, our method is able to find substantially higher-performance designs than either baseline method. On the power grid problem, our method finds designs that incur a higher best-case cost, since this problem includes a tradeoff between economic cost and robustness, but our method's designs are substantially more robust than the baselines, with much lighter tails in the cost distribution.

We observe that DR sometimes finds solutions that achieve lower average cost than those found by our method. We believe that this is due to DR optimizing against a less challenging failure population. This suggests the possibility of combining a failure dataset (predicted using our method) with an average-case dataset (sampled randomly from the prior) during repair; we hope to explore this and other adaptive strategies in future work.

**Benefits of differentiable simulation** Although we have designed our method to be compatible with either gradient-based or gradient-free MCMC subroutines, we observe that gradient-based samplers tend to converge faster than their gradient-free counterparts. Fig. 4 plots the performance of the best-performing design at each round against a static test set of 100 randomly sampled exogenous parameters for both gradient-based and gradient-free methods across all environments. Although these methods perform similarly on the formation problem, we see a clear pattern in the formation control, search-evasion, and power grid examples where gradient-based MCMC converges much faster, and this advantage is greater on higher-dimensional problems (second row), compensating for the additional time needed to compute the gradients (typically a 2-3x increase in runtime).

**Hardware experiments** We deploy the optimized hider and seeker trajectories in hardware using the Robotarium multi-robot platform [25] (we use 3 seekers and 5 hiders, since we had difficulty

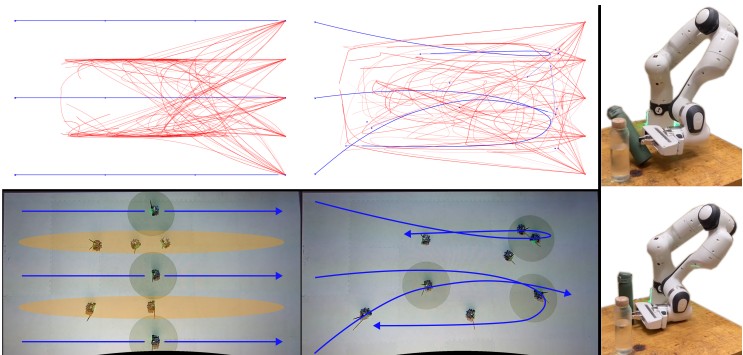

Figure 5: (Left) HW results for search-evasion with 5 hiders and 3 seekers, showing an initial search pattern (blue) and predicted failure modes (red). (Center) HW results for an optimized search pattern leaves fewer hiding places. (Right, top) An initial manipulation policy knocks over the object. (Right, bottom) The repaired manipulation policy pushes without knocking the bottle over.

testing with more agents in the limited space). We first hold the search pattern (design parameters) constant and optimize evasion patterns against this fixed search pattern, yielding the results shown on the left in Fig. 5 where the hiders easily evade the seekers. We then optimize the search patterns using our approach, yielding the results on the left where the hiders are not able to evade the seekers.

We also deploy an optimized policy for the pushing problem to a Franka Research 3 7-DoF robot arm. Fig. 5 shows a failure when the unoptimized policy fails to account for the uncertain center of mass of the bottle, as well as a successful execution with the repaired policy. Videos of all experiments are provided in the supplementary materials.

# 7 Discussion and Conclusion

Before sending any autonomous system out into the real world, it is important to understand how it will behave in range of operational conditions, including during potential failures. In this paper, we have presented a tool to allow the designers of autonomous systems to not only predict the ways in which a system is likely to fail but also automatically adjust their designs to mitigate those failures.

We apply our framework in simulation studies to a range of robotics and industrial control problems, including multi-robot trajectory planning and power grid control. Our results show that, relative to existing adversarial optimization methods, our novel sampling-based approach yields better predictions of possible failure modes, which in turn lead to more robust optimized designs. We also show empirically that, when it is possible to define a differentiable simulator, gradient-based MCMC methods allow our method to converge more than twice as fast as gradient-free methods.

## 7.1 Limitations

Since it would be prohibitively costly to search for failure cases in hardware experiments (especially if failures resulted in damage to the robot), our method is restricted to searching for failures in simulation. As such, it is limited to predicting only failures that are modeled by the simulator, excluding failures that could arise due to unmodeled effects. Practically, our method could be used in conjunction with hardware testing by catching some failures earlier in the development process and reducing the cost of eventual hardware testing.

A notable limitation of our approach is that it requires knowledge of the prior distribution of the exogenous disturbances $y$. Although this can be estimated in some cases (as in our experiments), in practice there may be uncertainty about the nature of this distribution. To address this, future works might investigate distributionally robust extensions of Algorithm 1 (akin to distributionally robust optimization methods [26]). Additional limitations are discussed in the appendix.

**Acknowledgments**

C. Dawson is supported by the NSF GRFP under Grant No. 1745302. This work was partly supported by the National Aeronautics and Space Administration (NASA) ULI grant 80NSSC22M0070, Air Force Office of Scientific Research (AFOSR) grant FA9550-23-1-0099, and the Defense Science and Technology Agency in Singapore. Any opinions, findings, and conclusions or recommendations expressed in this publication are those of the authors and do not necessarily reflect the views of the sponsors.

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

Table 1: Summary of notation

| | |
|---|---|
| $x \in \mathcal{X} \subseteq \mathbb{R}^{d_x}$ | Design parameters (controlled by system designer) |
| $y \in \mathcal{Y} \subseteq \mathbb{R}^{d_y}$ | Exogenous parameters (not controlled by designer) |
| $\xi \in \Xi \subseteq \mathbb{R}^{d_\xi}$ | Behavior of a system (e.g. the simulation trace) |
| $S : \mathcal{X} \times \mathcal{Y} \mapsto \xi$ | Simulator model of the system's behavior given design and exogenous parameters |
| $J : \Xi \mapsto R$ | Cost function |
| $J_r : \mathcal{X} \times \mathcal{Y} \mapsto R$ | Risk-adjusted cost function |
| $p_{x,0}(x), p_{y,0}(y)$ | Prior probability distributions for design and exogenous parameters |

## Summary of notation

Table 1 provides a summary of notation used in this paper.

## Sampling algorithms

Algorithm 1 relies on an MCMC subrouting for sampling from probability distributions given a non-normalized likelihood. Algorithms 2 and 3 provide examples of gradient-based (Metropolis-adjusted Langevin, or MALA) and gradient-free (random-walk Metropolis-Hastings, or RMH), respectively.

---

**Algorithm 2:** Metropolis-adjusted Langevin algorithm (MALA, [16, 22])

---

**Input:** Initial $x_0$, steps $K$, stepsize $\tau$, density $p(x)$.
**Output:** A sample drawn from $p(x)$.

1 **for** $i = 1, \ldots, K$ **do**
2      Sample $\eta \sim \mathcal{N}(0, 2\tau I)$        ▷ Gaussian noise
3      $x_{i+1} \leftarrow x_i + \tau \nabla \log p(x_i) + \eta$        ▷ Propose next state
4      $P_{accept} \leftarrow \frac{p(x_{i+1})e^{-||x_i - x_{i+1} - \tau \nabla \log p(x_{i+1})||^2/(4\tau)}}{p(x_i)e^{-||x_{i+1} - x_i - \tau \nabla \log p(x_i)||^2/(4\tau)}}$
5      With probability $1 - \min(1, P_{accept})$:
6          $x_{i+1} \leftarrow x_i$        ▷ Accept/reject proposal
7 **return** $x_K$

---

**Algorithm 3:** Random-walk Metropolis-Hastings (RMH, [23])

---

**Input:** Initial $x_0$, steps $K$, stepsize $\tau$, density $p(x)$.
**Output:** A sample drawn from $p(x)$.

1 **for** $i = 1, \ldots, K$ **do**
2      Sample $\eta \sim \mathcal{N}(0, 2\tau I)$        ▷ Gaussian noise
3      $x_{i+1} \leftarrow x_i + \eta$        ▷ Propose next state
4      $P_{accept} \leftarrow \frac{p(x_{i+1})e^{-||x_i - x_{i+1}||^2/(4\tau)}}{p(x_i)e^{-||x_{i+1} - x_i||^2/(4\tau)}}$
5      With probability $1 - \min(1, P_{accept})$:
6          $x_{i+1} \leftarrow x_i$        ▷ Accept/reject proposal
7 **return** $x_K$

---

## Proof of Theorem 5.1

We will show the proof for sampling from the failure distribution with likelihood given by Eq. (2); the proof for repair follows similarly. The log-likelihood for the failure distribution is

$$\log p_{y,0}(y) + J(x, y) \tag{4}$$

The authors of [16] show that MALA enjoys the convergence guarantees in Theorem 5.1 so long as the target log likelihood is strongly convex outside of a ball of finite radius $R$ (see Theorem 1 in [16]). More precisely, Ma et al. [16] give the bound

$$M \leq \mathcal{O}\left(e^{40LR^2} \frac{L^{3/2}}{(m-L)^{5/2}} d^{1/2} \left(d \ln \frac{L}{(m-L)} + \ln \frac{1}{\epsilon}\right)^{3/2}\right)$$

when step size is $\tau = \mathcal{O}\left(e^{-8LR^2}(m-L)^{1/2}L^{-3/2}\left(d\ln L/(m-L) + \ln 1/\epsilon\right)^{-1/2} d^{-1/2}\right)$.

Since $\log p_{y,0}(y)$ is assumed to be strongly $m$-convex, it is sufficient to show that as $||y|| \to \infty$, the strong convexity of the log-prior dominates the non-convexity in $J(x, y)$.

For convenience, denote $f(y) = J(x, y)$ and $g(y) = \log p_{y,0}(y)$. We must first show that $f(y) + g(y)$ is $(m-L)$-strongly convex, for which it suffices to show that $f(y) + g(y) - (m-L)/2||y||^2$ is convex. Note that

$$f(y) + g(y) - \frac{m-L}{2}||y||^2 = f(y) + \frac{L}{2}||y||^2 + g(y) - \frac{m}{2}||y||^2 \tag{5}$$

$g(y) - \frac{m}{2}||y||^2$ is convex by $m$-stong convexity of $g$, so we must show that the remaining term, $f(y) + L/2||y||^2$, is convex. Note that the Hessian of this term is $\nabla^2 f(y) + LI$. Since we have assumed that $J$ is $L$-Lipschitz smooth (i.e. its gradients are $L$-Lipschitz continuous), it follows that the magnitudes of the eigenvalues of $\nabla^2 f$ are bounded by $L$, which is sufficient for $\nabla^2 f(y) + LI$ to be positive semi-definite, completing the proof.

## AC Power Flow Problem Definition

The design parameters $x = (P_g, |V|_g, P_l, Q_l)$ include the real power injection $P_g$ and AC voltage amplitude $|V|_g$ at each generator in the network and the real and reactive power draws at each load $P_l, Q_l$; all of these parameters are subject to minimum and maximum bounds that we model using a uniform prior distribution $p_{x,0}$. The exogenous parameters are the state $y_i \in \mathbb{R}$ of each transmission line in the network; the admittance of each line is given by $\sigma(y_i)Y_{i,nom}$ where $\sigma$ is the sigmoid function and $Y_{i,nom}$ is the nominal admittance of the line. The prior distribution $p_{y,0}$ is an independent Gaussian for each line with a mean chosen so that $\int_{-\infty}^{0} p_{y_i,0}(y_i)dy_i$ is equal to the likelihood of any individual line failing (e.g. as specified by the manufacturer; we use 0.05 in our experiments). The simulator $\mathcal{S}$ solves the nonlinear AC power flow equations [27] to determine the state of the network, and the cost function combines the economic cost of generation $c_g$ (a quadratic function of $P_g, P_l, Q_l$) with the total violation of constraints on generator capacities, load requirements, and voltage amplitudes:

$$J = c_g + v(P_g, P_{g,min}, P_{g,max}) + v(Q_g, Q_{g,min}, Q_{g,max}) \tag{6}$$
$$+ v(P_l, P_{l,min}, P_{l,max}) + v(Q_l, Q_{l,min}, Q_{l,max}) \tag{7}$$
$$+ v(|V|, |V|_{min}, |V|_{max}) \tag{8}$$

where $v(x, x_{min}, x_{max}) = L\left([x - x_{max}]_+ + [x_{min} - x]_+\right)$, $L$ is a penalty coefficient ($L = 100$ in our experiments), and $[\circ]_+ = \max(\circ, 0)$ is a hinge loss.

Efficient solutions to SCOPF are the subject of active research [28] and an ongoing competition run by the U.S. Department of Energy [29]. In addition to its potential economic and environmental impact [27], SCOPF is also a useful benchmark problem for 3 reasons: 1) it is highly non-convex, 2) it has a large space of possible failures, and 3) it can be applied to networks of different sizes

to test an algorithm's scalability. We conduct our studies on one network with 14 nodes and 20 transmission lines (32 design parameters and 20 exogenous parameters) and one with 57 nodes and 80 lines (98 design parameters, 80 exogenous parameters)

The simulator $S$ solves the nonlinear AC power flow equations [5, 30] for the AC voltage amplitudes and phase angles ($|V|, \theta$) and the net real and reactive power injections ($P, Q$) at each bus (the behavior $\xi$ is the concatenation of these values). We follow the 2-step method described in [30] where we first solve for the voltage and voltage angles at all buses by solving a system of nonlinear equations and then compute the reactive power injection from each generator and the power injection from the slack bus (representing the connection to the rest of the grid). The cost function $J$ is a combination of the generation cost implied by $P_g$ and a hinge loss penalty for violating constraints on acceptable voltages at each bus or exceeding the power generation limits of any generator, as specified in Eq. 8. The data for each test case (minimum and maximum voltage and power limits, demand characteristics, generator costs, etc.) are loaded from the data files included in the MAT-POWER software [31].

This experiment can be run with the `solve_scacopf.py` script in the `experiments/power_systems` directory.

## Search-Evasion Problem Definition

This problem includes $n_{seek}$ seeker robots and $n_{hide}$ hider robots. Each robot is modeled using single-integrator dynamics and tracks a pre-planned trajectory using a proportional controller with saturation at a maximum speed chosen to match that of the Robotarium platform [25]. The trajectory $\mathbf{x}_i(t)$ for each robot is represented as a Bezier curve with 5 control points $\mathbf{x}_{i,j}$,

$$\mathbf{x}_i(t) = \sum_{j=0}^{4} \binom{4}{j} (1-t)^{4-j} t^j \mathbf{x}_{i,j}$$

The design parameters are the 2D position of the control points for the trajectories of the seeker robots, while the exogenous parameters are the control points for the hider robots. The prior distribution for each set of parameters is uniform over the width and height of the Robotarium arena ($3.2\,\mathrm{m} \times 2\,\mathrm{m}$).

We simulate the behavior of the robots tracking these trajectories for $100\,\mathrm{s}$ with a discrete time step of $0.1\,\mathrm{s}$ (including the effects of velocity saturation that are observed on the physical platform), and the cost function is

$$J = \sum_{i=1}^{n_{hide}} \left( \widetilde{\min_{t=t_0,\ldots,t_n}} \left( \widetilde{\min_{j=1,\ldots,n_{seek}}} \left\| \mathbf{p}_{hide,i}(t) - \mathbf{p}_{seek,j}(t) \right\| - r \right) \right)$$

where $r$ is the sensing range of the seekers ($0.5\,\mathrm{m}$ for the $n_{seek} = 2$ case and $0.25\,\mathrm{m}$ for the $n_{seek} = 3$ case); $\widetilde{\min}(\circ) = -\frac{1}{b}\mathrm{logsumexp}(-b \circ)$ is a smooth relaxation of the element-wise minimum function where $b$ controls the degree of smoothing ($b = 100$ in our experiments); $t_0, \ldots, t_n$ are the discrete time steps of the simulation; and $\mathbf{p}_{hide,i}(t)$ and $\mathbf{p}_{seek,j}(t)$ are the $(x, y)$ position of the $i$-th hider and $j$-th seeker robot at time $t$, respectively. In plain language, this cost is equal to the sum of the minimum distance observed between each hider and the closest seeker over the course of the simulation, adjusted for each seeker's search radius.

This experiment can be run with the `solve_hide_and_seek.py` script in the `experiments/hide_and_seek` directory.

## Formation Control Problem Definition

This problem includes $n$ drones modeled using double-integrator dynamics, each tracking a pre-planned path using a proportional-derivative controller. The path for each drone is represented as a Bezier, as in the pursuit-evasion problem.

The design parameters are the 2D position of the control points for the trajectories, while the exogenous parameters include the parameters of a wind field and connection strengths between each pair of drones. The wind field is modeled using a 3-layer fully-connected neural network with $\tanh$ saturation at a maximum speed that induces $0.5\,\mathrm{N}$ of drag force on each drone.

We simulate the behavior of the robots tracking these trajectories for $30\,\mathrm{s}$ with a discrete time step of $0.05\,\mathrm{s}$, and the cost function is

$$J = 10||COM_T - COM_{goal}|| + \max_t \frac{1}{\lambda_2(q_t) + 10^{-2}}$$

where $COM$ indicates the center of mass of the formation and $\lambda_2(q_t)$ is the second eigenvalue of the Laplacian of the drone network in configuration $q_t$. The Laplacian $L = D - A$ is defined in terms of the adjacency matrix $A = \{a_{ij}\}$, where $a_{ij} = s_{ij}\sigma\left(20(R^2 - d_{ij}^2)\right)$, $d_ij$ is the distance between drones $i$ and $j$, $R$ is the communication radius, and $s_{ij}$ is the connection strength (an exogenous parameter) between the two drones. The degree matrix $D$ is a diagonal matrix where each entry is the sum of the corresponding row of $A$.

This experiment can be run with the `solve.py` script in the `experiments/formation2d` directory.

## F16 GCAS Problem Definition

This problem is based on the ground collision avoidance system (GCAS) verification problem originally posed in [32], where the challenge is to design a controller for an F16 jet aircraft that avoids collision with the ground when starting from a range of initial conditions. We use the JAX implementation [33] of the original F16 dynamics model published in [32]. This model has 15 states and 4 control inputs, and it includes a nonlinear engine model and an approximate aerodynamics model. The original model was published with a reference GCAS controller that is not able to maintain safety over the desired range of initial conditions (given in Table 4 in [32]). We supplement this reference controller with a neural network controller that only activates below a specified altitude threshold; the parameters of this network and the value of the altitude threshold represent our design parameters (approximately 1,800 total parameters, with a uniform prior over these parameters). The exogenous parameters include the initial altitude, roll, pitch, roll rate, and pitch rate $(h, \phi, \theta, p, q)$, drawn from Gaussian distributions:

$$h \sim \mathcal{N}(1500, 200)\ ft$$
$$\phi \sim \mathcal{N}(0, \pi/8)$$
$$\theta \sim \mathcal{N}(-\pi/5, \pi/8)$$
$$p \sim \mathcal{N}(0, \pi/8)$$
$$q \sim \mathcal{N}(0, \pi/8)$$

The cost function is

$$J = [200 - \min h]_+/10 + \frac{1}{T}\sum_{t=1,\ldots,T}\left[\left(\frac{\phi}{\pi}\right)^2 + \left(\frac{\theta}{\pi}\right)^2 + \left(\frac{\alpha}{\pi}\right)^2 + \left(\frac{\beta}{\pi}\right)^2\right]$$

where $h$ is altitude in feet, $\phi$ is roll, $\theta$ is pitch, $\alpha$ is angle of attack, $\beta$ is sideslip angle, $[\cdot]_+$ is the exponential linear unit and $\min$ is a soft `log-sum-exp` minimum. Empirically, $J \geq 15$ indicates that the aircraft has crashed ($h = 0$). The simulation is run for $15\,\mathrm{s}$ with timestep $0.01\,\mathrm{s}$ but stopped early if the aircraft crashes or leaves the flight envelope where the aerodynamic model is accurate ($-10° \leq \alpha \leq 45°$ and $|\beta| \leq 30°$).

## Pushing Problem Definition

In this problem, we model the task of pushing an obstructing object out of the way so that the robot can grasp another object. The robot receives noisy observations of the height, radius, and position of

the obstructing object and uses this information to plan a pushing action (push height and force) that can move the object to the side. The robot must ensure, without knowing the frictional or inertial properties of the object, that its push is just forceful enough to move the object without knocking it over.

The design parameters include 1,200 parameters of a neural network used to predict push height and force given noisy observations about the object. The exogenous parameters include the true height $h$, true radius $r$, mass $m$, center-of-mass height $h_{com}$, and friction coefficient $\mu$ between the object and the ground, plus the noisy observations of the height and radius $\hat{h}$ and $\hat{r}$. We use a uniform parameters over the design parameters and the following priors for the exogenous parameters:

$$h \sim U(0.1, 0.2) \; m$$
$$r \sim U(0.1, 0.25) \; m$$
$$m \sim U(0.1, 1.0) \; kg$$
$$\mu \sim U(0.1, 1.0)$$
$$h_{com} \sim U(0.1h, 0.9h)$$
$$\hat{h} \sim \mathcal{N}(h, 0.1)$$
$$\hat{r} \sim \mathcal{N}(r, 0.1)$$

The cost function is

$$J = [\sum \tau]_+ + [1 - \sum F]_+$$

where $[\cdot]_+$ is the ReLU function, $\sum \tau$ is the net moment applied to the object about its tipping point (defined as positive in the direction of tipping, so that negative moments are stable), and $\sum F$ is the net force in the pushing direction.

The simulator that we use in this example is fairly simple. It models the object as a cylinder resting on a flat plane, and we make the assumption that a successful push (i.e. no tipping) is quasi-static. With this assumption, we can detect a successful push by computing the net force and moment applied to the object (including static friction). The object will tip (failure) if the net moment about the edge of the cylinder is positive, and the object will not move (also failure) if the net force is zero. The point of this experiment is to show how even a simplified model can predict and repair certain failures, and that these repairs can transfer to hardware. There are certainly failures that this simulator will not catch (e.g. surface irregularities in the table might catch the edge of the object and cause it to flip), and there are false positives that our simulator will detect (e.g. a near-failure where the transition from static to dynamic friction reduces the net moment and prevents tipping), but we hope that this shows that our method can still be useful with a low-fidelity model.

## Hyperparameters

Table 2 includes the hyperparameters used for each environment.

## Details on Hardware Experiments

### Search-evasion

The search-evasion hardware experiment was implemented on the Robotarium platform, an open-access multi-agent research platform [25]. Trajectories for the hiders and seekers were planned offline using our method (with $K = 100$ rounds and $M = 10$ substeps per round, taking $41\,\mathrm{s}$) and then tracked online using linear trajectory-tracking controllers.

### Pushing

The pushing experiment was implemented using a Franka Research 3 7-DOF robot arm and an Intel RealSense D415 RGBd camera. The camera was positioned over the robot's workspace, and

the depth image was used to segment the target and obstructing objects, as well as estimating the height and radius of each object. These estimates were passed to the planning neural network, using weights trained using our method ($K = 100$, $M = 10$). The planning network predicts a push height and force; the pushing action is executed using a Cartesian impedance tracking controller.

## Further Limitations

In this paper, we restrict our attention to problems with continuous design and exogenous parameters, since gradient-based inference methods realize the greatest benefit on problems with a continuous domain. It is possible to extend our method to problems with mixed continuous-discrete domains by using a gradient-based sampling algorithm for the continuous parameters and a gradient-free sampler for the discrete parameters; we hope to explore the performance implications of this combination in future work.

Although the sampling methods we use in this paper remain theoretically sound when the simulator and cost landscape are discontinuous (as is the case in manipulation problems, for example), it remains to be seen what practical effects discontinuity might have on convergence rate and solution quality.

Finally, although we include details on how to use tempering with our approach, we found that tempering was not needed for convergence on any of the problems we studied; more work is needed to understand when tempering is necessary for convergence of MCMC-based algorithms on various robotics problems.

Table 2: Hyperparameters used for each environment. $n_q$ is the number of quenching rounds; $i$ denotes the round number in Alg. 1.

| Environment | $n_x$ | $n_y$ | $\tau$ | $K$ | $M$ | $n_q$ | $\lambda$ |
|---|---|---|---|---|---|---|---|
| Formation (5 agents) | 5 | 5 | $10^{-3}$ | 50 | 5 | 5 | $e^{-5i}$ |
| Formation (10 agents) | 5 | 5 | $10^{-3}$ | 50 | 5 | 5 | $e^{-5i}$ |
| Search-evasion (6 seekers, 10 hiders) | 10 | 10 | $10^{-2}$ | 100 | 10 | 25 | $e^{-5i}$ |
| Search-evasion (12 seekers, 20 hiders) | 10 | 10 | $10^{-2}$ | 100 | 10 | 25 | $e^{-5i}$ |
| Power grid (14-bus) | 10 | 10 | $10^{-6}$ for $x$ $10^{-2}$ for $y$ | 100 | 10 | 10 | $e^{-5i}$ |
| Power grid (57-bus) | 10 | 10 | $10^{-6}$ for $x$ $10^{-2}$ for $y$ | 100 | 10 | 10 | $e^{-5i}$ |
| F16 | 10 | 10 | $10^{-2}$ for $x$ $10^{-4}$ for $y$ | 100 | 5 | 0 | $e^{-10i}$ |
| Pushing | 10 | 10 | $10^{-2}$ | 10 | 10 | 0 | $e^{-10i}$ |

