# OpenReview forum: "A Bayesian approach to breaking things: efficiently predicting and repairing failure modes via sampling"
_robot-learning.org/CoRL/2023/Conference — CoRL 2023 Poster_

### Official Review · Reviewer_fucE · 2023-07-08

**Confidence:** 3
**Originality:** Good
**Technical Quality:** Good
**Clarity Of Presentation:** Good
**Impact:** 4

**Recommendation:**

Weak Accept: I recommend accepting the paper, but will not argue for my recommendation if the majority of other reviewers have a different opinion.

**Review:**

This is not the typical sort of paper you see at CoRL, but i thought it was an interesting idea and the paper presentation was nice and clear.

**Pseudo-posteriors / likelihoods.**
I think this paper would be improved by framing equations 2 and 3 as 'pseudo'-posteriors. These methods are seen across machine learning under different names, and I think pseudo-posteriors captures the Bayesian inference aspect while acknowledging there is no proper likelihood function. Pseudo posteriors are derived from KL regularized optimization problem, so they are a nice way of framing optimization-as-inference methods. Pseudo posteriors also make the 'tempering' problem explicit, since the temperature is part of the objective. The paper only mentions the temperature tuning problem in passing.

Some relevant references

O. Catoni. Statistical learning theory and stochastic optimization

On the properties of variational approximations of Gibbs posteriors
P Alquier, J Ridgway, N Chopin - The Journal of Machine Learning 2016

A. S. Dalalyan and A. B. Tsybakov. Aggregation by exponential weighting, sharp PAC-
Bayesian bounds and sparsity

Inferring Smooth Control: Monte Carlo Posterior Policy Iteration with Gaussian Processes
J Watson, J Peters - Conference on Robot Learning, 2023

**Different approximate inference methods.** I would have liked to have seen some more exploration of different approximate inference techniques, especially since JAX was used. I imagine methods like stein variational gradient descent, the Laplace approximation and automatic differentiation VI would be straight forward to integrate with the JAX code.

**Sim-to-real RL.** It would have been nice to have seen a sim-to-real deep RL experiment to assess that this method can scale.  I imagine this would need a more scalable approximate inference method and probably some integration with entropy regularized RL.

**Experiment seeds and uncertainty intervals.**
Figure 4 is missing different random seed runs and uncertainty intervals. It also looks like Figure 3 has many random test cases but only for a single run? rather than different seeds? I would have also been good if Figure 4 captured the learning speed of the baselines,


**Quality Of The Limitations Section:**

Limitations are addressed clearly

**Questions For Rebuttal:**

1) How well did MALA perform as an optimizer compared to DR and GD? Figure 3 combines different optimization methods and robustness strategies, so its hard to know what is significant. Is the poor performance in 3c) and f) due to the robustness being too strong or the optimizer finding worse minima?

2) What role did the temperature play in the optimization? How did the schedule need to be tuned?

**Robotics Focus:**

Sufficient demonstration on hardware

**Summary Of Paper:**

The paper uses optimization-as-inference in an adversarial setting to optimize against worst-case failure modes.
This is done using MCMC methods.
The method is evaluated on a multi-agent hide-and-seek task and power network control.

**Summary Of Recommendation:**

On the whole I liked the idea and it felt like an interesting way of doing robustness and optimization-as-inference.

---

### Official Review · Reviewer_nExT · 2023-07-18

**Confidence:** 4
**Originality:** Good
**Technical Quality:** Good
**Clarity Of Presentation:** Very Good
**Impact:** 3

**Recommendation:**

Weak Accept: I recommend accepting the paper, but will not argue for my recommendation if the majority of other reviewers have a different opinion.

**Review:**

Simultaneous stress-testing and improvement of autonomous systems is an interesting topic, and I do like the idea of leveraging Bayesian inference for achieving this. But, I have some reservations about the algorithm presented in this paper:
 1. **Prior Selection:** The approach is sensitive to the choice of the priors, used in lines 4 and 7 of algorithm 1. A good prior would be very difficult to prescribe for a complex autonomous system. The paper does not discuss in much detail how a prior can be chosen. The experimental section also does not provide any prior selection details for the presented examples.
2. **Catastrophic Forgetting:** The design distribution $p_{x,i}$ at iteration $i$ is based on the failure samples from the previous iteration $[y]_{i-1}$. How is it ensured that the new designs are still able to address the failure modes from all iterations prior to $i-1$?
3. **Simulator:** As the approach relies on a simulator, how do the authors envision addressing the inevitable sim-to-real gap? Additionally, I don't completely understand what Assumption 1 (almost everywhere differentiable simulator) entails.
4. **Bayesian RL**: Another aspect that I would like to understand better is how this algorithm compares to Bayesian RL approaches. I see some strong parallels, for instance failure prediction can be viewed as exploration while design parameter selection can be viewed as exploitation.
5. **Experimental Evaluation:** Although the environments presented in the paper demonstrate the algorithm, they seem more academically inclined and do not demonstrate the algorithm's real-world significance. Furthermore, the third environment, power grid dispatch, is completely disconnected from robotics. It would have been great to see some application where a robot's autonomy stack is stress tested for failures and then repaired using this algorithm.

_Edit after rebuttal_

I am still trying to convince myself that the paper has sufficient novelty. I viewed the ability of this paper to provide useful insights about what is breaking the system to be a big draw. Turns out that is not really the case. Without it, I am having a hard time understanding how this approach stands apart from the adversarial attack literature in ML where it is fairly common to search for examples that result in failures and then training with them to robustify the network. Indeed, there has been work on attacks with Bayesian optimization as well.
However, the authors did a stellar job during the rebuttal phase and have introduced a hardware demo for their approach along with a theoretical extension, so I am adjusting my score to Weak Accept.

**Quality Of The Limitations Section:**

Additional details required

**Questions For Rebuttal:**

Please provide clarifications for my comments above.

**Robotics Focus:**

Relevant but unlikely to deploy to hardware in near future

**Summary Of Paper:**

This paper presents an approach for predicting failures for autonomous systems and then repairing the failures by re-designing the system. Failure prediction and repair are achieved by Bayesian inference.

**Summary Of Recommendation:**

My current recommendation is a Weak Reject because the presented approach suffers from a few challenges that make me wonder about its scalability beyond academic examples.

_Edit after rebuttal_

Score updated to Weak Accept.

---

### Official Review · Reviewer_GT1s · 2023-07-18

**Confidence:** 3
**Originality:** Good
**Technical Quality:** Very Good
**Clarity Of Presentation:** Very Good
**Impact:** 3

**Recommendation:**

Weak Accept: I recommend accepting the paper, but will not argue for my recommendation if the majority of other reviewers have a different opinion.

**Review:**

The paper is very well written, ideas are conveyed concisely and it is very easy to follow along. The contributions of the paper are stated clearly and explained well.

While I am not entirely familiar with the previous literature on failure mode prediction, it seems that the main ideas of the paper are novel and constitute a meaningful contribution to the field of autonomous system verification.

The experiments presented in the paper showcase the generality of this method by demonstrating its usability on a variety of robotic domains, some of which may have real-world impact.

Despite its versatility displayed by the tested tasks, I am concerned that the usability of this approach to real world robotic scenarios is limited to those where good simulators are available.

All of the domains shown in the paper enjoy simulators which model both the design parameters and the failure modes well. However, for many robotic tasks (such as object manipulation or quadrupedal navigation) designing good simulators which can predict failure modes well is an extremely complex task: consider, for instance, failures arising from friction between objects.

Another concern is related to the complexity of the computation required for the sampling procedure. The experiment shown on real hardware is relatively small, with low-dimensional spaces for both the design and exogenous parameter spaces. While setting design parameters offline for a single scenario may work for many domains, calculating trajectories repeatedly may be computationally prohibitive, depending on the dimensionality of the problem.

I believe an important addition to this paper is a discussion of the types of tasks and domains to which the approach proposed in the paper is applicable. At the very least, this issue should be mentioned in the limitations section.

**Quality Of The Limitations Section:**

Limitations are addressed clearly

**Questions For Rebuttal:**

Are there any restrictions on the structure of the distributions $p_{x,i}, p_{y,i}$? While they should clearly allow easy sampling and inference of likelihood, would the proposed method be compatible with high-dimensional, complex distributions such as ones represented by deep neural networks?

Some details are unclear regarding the real hardware experiment - is it run for two iterations of only (i.e. $K=2$)? Or does it use trajectories obtained from the simulated hide and seek environment?

While it is not the focus of the paper, it is interesting to note that in some of the tasks, the DR approach performs better on average than the proposed algorithm on the sampled test set of exogenous parameters (Figure 3). While it makes sense that DR is not optimizing against challenging failure cases and therefore performs better, is there a way to avoid the loss of performance in the average case while optimizing for the worst case using the authors’ proposed method?

Regarding convergence rates (Figure 4): it is clear that the gradient-based sampling method converges faster in terms of sampling rounds. However, how long does it take in practice to perform these iterations, and how does the dimensionality of the problem affect this duration?

**Robotics Focus:**

Sufficient demonstration on hardware

**Summary Of Paper:**

This paper proposes a reframing of adversarial optimization as a Bayesian inference problem, by sampling system control parameters and failure modes from probability distributions. The proposed approach relies on the availability of differentiable simulators describing both the control behavior of the autonomous systems in question, as well as the exogenous parameters affecting the states of the agents. The paper demonstrates this approach on three different simulated domains, as well as a simple experiment on a real-world environment.

**Summary Of Recommendation:**

While the paper proposes a simple and novel idea and demonstrates its usability on multiple simulated domains, some concerns are present regarding usability to real-world robotic scenarios. Were I to be convinced of the applicability and scalability of this approach, I would be happy to raise my score accordingly.

EDIT: The authors have addressed my concerns in their rebuttal, and have spent considerable time and effort to improve their paper in the given time, including an additional hardware experiment. I believe the updated version of the paper would be a valuable addition to the robot learning community. Therefore, I have updated my score to "weak accept".

---

### Official Review · Reviewer_meR7 · 2023-07-25

**Confidence:** 4
**Originality:** Fair
**Technical Quality:** Fair
**Clarity Of Presentation:** Good
**Impact:** 2

**Recommendation:**

Strong Reject: I recommend rejecting the paper and will argue for my recommendation even if other reviewers hold a different opinion.

**Review:**

Strengths:
1. The paper's comprehensive evaluation on diverse real-world robotics and control problems provides valuable insights into the method's practical applicability and versatility.

Weaknesses:
1. The methodological contribution appears limited, as the MCMC algorithm and the proposed sampling or gradient-based optimization techniques are well-known and seem to be straightforward applications of standard optimization approaches.
2. The absence of theoretical results and analysis leaves important questions unanswered, such as the algorithm's convergence, optimality, and the underlying assumptions regarding exogenous parameters.
3. The reliance on strong assumptions, such as the assumed known cost function and explicit knowledge of the distribution of x and y variables, raises concerns about the method's generalizability and adaptability to various real-world scenarios, particularly in settings with uncertain or incomplete information.
4. The fairness of the comparison in the experiments is questionable. The advantage provided by known distributions in the proposed method, compared to the adversarial optimization approach, may skew the results in favor of the former, potentially affecting the scalability and performance assessment.

**Quality Of The Limitations Section:**

Limitations are addressed clearly

**Questions For Rebuttal:**

1. Can you clarify the novel aspects of the Bayesian inference-based method and how it extends beyond conventional optimization approaches?
2. How do you plan to provide theoretical insights on the algorithm's convergence, optimality, and stability under specific assumptions?
3.  How will you address concerns about the method's reliance on strong assumptions, such as the known cost function and the distribution of x and y variables, to ensure adaptability in scenarios with uncertain or incomplete information?
4. How will you ensure a more balanced and unbiased comparison with other optimization approaches, considering the advantage provided by known distributions in the proposed method?

**Robotics Focus:**

Sufficient demonstration on hardware

**Summary Of Paper:**

The paper presents a Bayesian inference-based method capable of automatically predicting potential system failures and generating corresponding repair solutions. The proposed algorithms for automatic failure detection and repair are designed in an evolutionary optimization manner, utilizing both gradient-based and sampling-based methods to find solutions. Extensive testing on three real-world robotics and control problems demonstrates promising results in terms of speed and performance.


**Summary Of Recommendation:**

the paper introduces an interesting approach to automatic failure detection and repair in robotics and control using Bayesian inference. However, addressing the aforementioned weaknesses is crucial to enhancing the paper's scientific contribution and practical relevance. Providing theoretical insights, addressing assumptions, and conducting more robust and fair comparisons would significantly strengthen the paper and make it a valuable addition to the field of automatic failure detection and repair in robotics and control. As such, I recommend rejection for the current version.

---

### Decision · Program_Chairs · 2023-08-30

**Decision:**

Accept (Poster)

**Comment:**

This paper frames the problem of discovering failure modes of a system as optimization with uncertain variables. A stochastic optimization problem is formulated, and solved using a form of coordinate descent (for the design parameters and the adversarial disturbance parameters), and each coordinate optimization step is done by casting the optimization problem as an inference problem, and using MCMC.

The initial reviews were mostly negative, with concerns about the novelty of the approach, the soundness of the methodology, the relevance of the experiments, and the relevance to CoRL. The authors managed to pull off a commendable rebuttal, including some theoretical analysis and real robot results on simple object manipulation. Following the rebuttal, most reviewers expressed a positive view of the paper, and agreed to accept it.

I find the empirical finding that the MCMC variants work well in this problem formulation on a variety of tasks to be interesting, and therefore I agree with the reviewers and recommend acceptance.
However, the theoretical analysis feels rushed. The current results are for optimizing each coordinate (x and y), but there is no result on the complete optimization - what guarantees that the optimization doesn’t oscillate? The authors should clearly discuss this in their final version of the paper.